# A Walk in the Park? Examining the Impact of App-Based Weather Warnings on Affective Reactions and the Search for Information in a Virtual City

**DOI:** 10.3390/ijerph18168353

**Published:** 2021-08-06

**Authors:** Samuel Tomczyk, Maxi Rahn, Henriette Markwart, Silke Schmidt

**Affiliations:** Department Health and Prevention, Institute of Psychology, University of Greifswald, Robert-Blum-Straße 13, 17489 Greifswald, Germany; maxi.rahn@uni-greifswald.de (M.R.); henriette.markwart@uni-greifswald.de (H.M.); silke.schmidt@uni-greifswald.de (S.S.)

**Keywords:** emergency alert, public warning systems, affect heuristic, weather, disaster, online experiment

## Abstract

*Background*: Warning apps can provide personalized public warnings, but research on their appraisal and impact on compliance is scarce. This study introduces a virtual city framework to examine affective reactions when receiving an app-based warning, and subsequent behavioral intentions. *Methods:* In an online experiment, 276 participants (M = 41.07, SD = 16.44, 62.0% female) were randomly allocated to one of eight groups (warning vs. no warning, thunderstorm vs. no thunderstorm, video vs. vignette). Participants were guided through a virtual city by a mock-up touristic app (t1). Then, the app issued a warning about an impending thunderstorm (t2), followed by a virtual thunderstorm (t3). The virtual city tour was presented via vignettes or videos. ANCOVAs were used to investigate trajectories of momentary anxiety, hierarchical regressions analyzed the impact of momentary anxiety on information seeking. *Results*: Participants who received a warning message and were confronted with a thunderstorm showed the highest increase in momentary anxiety, which predicted information seeking intentions. *Conclusions*: The findings underscore the importance of affective appraisal in processing warning messages. The virtual city framework is able to differentiate the impact of warning versus event in an online context, and thus promising for future warning research in virtual settings.

## 1. Introduction

Severe weather comes in a variety of forms, such as droughts, floods, or thunderstorms, each of which may have massive societal and meteorological impact [1]. The World Meteorological Organization has developed a definition of severe weather that acknowledges regional differences in types of (e.g., sand/snow storms, flash floods) as well as thresholds for severe weather events (e.g., intensity of wind or rain) but characterizes severe weather as “*an extreme meteorological event or phenomenon, which represents a real hazard (to human life and property)*” ([2], p. 2). Although severe weather is less intense and occurs less frequently in the European region than in other parts of the world, it still poses an increasing threat in Europe [3,4]: for instance, thunderstorms, or so-called convective storms, can arise with heavy winds or even tornadoes, rainfall, hail, and lightning [5,6,7,8,9].

By alerting and providing information to populations that are potentially affected by thunderstorms and other large-scale weather events, warning messages are an effective means of preventing various forms of damage, including loss of life and property [9,10]. A large body of literature informs the construction of effective warning messages from an interdisciplinary perspective [10,11]. For example, studies on weather warnings indicate that warning messages emphasizing potential consequences and providing guidance for protective measures are perceived as more threatening and are more likely to be followed than those, which describe the event alone [12,13].

Traditionally, TV and radio news broadcast weather reports and weather warnings, but in recent years, new channels have been utilized to communicate warning messages. Mobile warning apps, such as FEMA in the U.S., or NINA and KATWARN in Germany, are able to promptly alert a broad range of the population regarding hazards and threats while simultaneously providing them with information on protective measures [14,15,16]. While some apps inform about a range of hazards (e.g., FEMA and KATWARN), others specifically focus on weather forecasting and severe weather (e.g., WarnWetter, a weather app), with the latter being highly popular [17,18]. A survey conducted among university students found that around 80% used weather warning apps at least once per day to get forecasts, thus indicating high use rates compared to traditional media, such as local television (6.8%) [19]. Yet, information processing of app-based warning messages has not been conclusively investigated, for instance, a range of new requirements needs to be considered (e.g., brevity of the warning message or format of the warning). As they play a seminal role in the future of civil protection regarding environmental hazards, it is important to comprehensively understand how mobile warning messages are processed by recipients.

Sutton and Kuligowski [20] provide an overview of existing short messaging channels used to communicate warning messages and they outline a theoretical background referring to an adaption of the Protective Action Decision Model (PADM) [21,22]. The PADM describes information processing and human responses towards hazards and threats through several stages, such as pre-decisional processes and core perceptions, followed by decision-making processes. Social or environmental cues, as well as information provided by warning messages, and a recipient’s characteristics, can elicit pre-decisional processes, which consist of the reception, attention, and comprehension of said cues or information. Consequently, an estimation of credibility and personal threat or risk can take place. If a cue is perceived as credible and one’s own risk is perceived as high (including cognitive and affective appraisal of the threat), the individual will seek protective actions and implement them. This makes risk appraisal an important factor in the warning process.

### 1.1. Risk Appraisal and Warning Compliance

However, several studies revealed complex relationships between risk appraisal and behavior, in that perceived risk is not necessarily accompanied by the implementation of protective measures [23,24,25]. One reason for this could be the varying—and partly one-sided–cognitive operationalization of risk appraisal [26,27]. Beyond cognitive risk appraisal, research from the field of health behavior [28] and on different hazards and warnings [29,30,31,32] showed that affective responses are as important when examining hazard-related information processing and compliance. The Protection Motivation Theory [33] and the Extended Parallel Process Model [34], for example, formalize this distinction in the context of fear appeals. They discern threat appraisals and efficacy appraisals as key determinants of compliance, with negative affective reactions (e.g., fear) leading to higher threat appraisal (i.e., affective risk appraisal), and positive evaluations of personal coping capabilities (e.g., self-efficacy) leading to more positive efficacy beliefs. Taken together, high threat perceptions and high efficacy beliefs predict behavioral compliance with a warning message. Conversely, a study on home fires showed that positive affective associations with wood heating (i.e., lower affective risk appraisal) attenuated the effects of cognitive risk appraisal on using alternative heating systems, thus weakening the association between risk perceptions and behavior [35].

Another theoretical model considering such affective reactions towards threats is the affect heuristic, which differentiates between an analytic and an experiential system when it comes to information processing [36,37,38]. The analytical system describes a slower way of processing information, which is activated when there is enough time to weigh all the facts and information about the situation. In contrast, the experiential system is driven by affective reactions and emotions towards the hazard. It is activated when there is an acute threat, so that judgments and decisions have to be made fast or under time pressure. In this sense, warning messages represent acute threats, as they point to imminent danger. This perspective aligns with neurobiological assumptions based on the BIS/BAS Model [39,40,41,42], which postulates two motivational systems—the BIS (behavioral inhibition) and the BAS (behavioral approach)—that influence affect and behavior. While BAS is activated by cues of reward and may enhance approach tendencies, BIS is sensitive towards cues of punishment and threat, and therefore often associated with avoidance tendencies. When confronted with an unknown or threatening cue—for example after receiving a warning message for a thunderstorm or perceiving other thunderstorm-related stimuli, such as dark clouds or growling thunder–activation of the BIS can increase attention towards the cue and enhance arousal and negative emotions such as anxiety. This in turn can lead to the inhibition or reduction of current behaviors, eventually followed by avoidance behavior. With the BIS activated, a person may interrupt their current activity, collect more information about the impending threat, and implement protective measures.

To date, applied warning research has produced ample evidence for these assumptions: For instance, a study on pop-up messages showed that negative affect impacts the perception and processing of relatively trivial threats (e.g., outdated software while surfing the internet) [43]. Meta-analyses and studies of warning messages on tobacco products highlighted the advantages of pictorial warnings over text-based warnings by evoking more negative emotional responses (e.g., fear, sadness, or disgust), as well as higher intentions not to start or to quit smoking. Thus, it appears that warnings on tobacco products influence behavioral intentions by increasing negative affect [44,45,46]. Similar affective response patterns also seem to play a role in weather-related hazards. For example, exposure-based research in a clinical sample found that the (virtual) exposure to a threatening stimulus can elicit affective, mainly fear-related reactions [47,48,49]. These effects are applied in exposure-based cognitive behavioral therapies for storm-related phobias [50]. Previous research on other hazards like earthquakes [51], fires [52], and floods [53] has also used virtual reality to create a realistic scenario that can induce negative affect when confronted with an event. This allows researchers to study and modify subsequent affective, cognitive, and behavioral reactions.

Consequently, eliciting negative affect could also be a key factor of weather warning compliance in that increased fear will lead to increased compliance intentions or behaviors [30,31,54]. Nevertheless, receiving warning messages can also lead to a reduction in negative affect [29]. A meta-analytic study of fear appeal theory offers a potential explanation for this countervailing trend, namely an interaction between threat and efficacy [55]. The receipt of a warning message could increase salience of a threat, thereby evoking negative affect. But simultaneously communicating effective protective measures against the threat could lead to increased efficacy, which, in turn, might reduce negative affect.

### 1.2. Research Questions and Hypothesis

Despite recommendations to investigate mobile warning messages under conditions that are as realistic as possible [11,56], few studies have investigated said processes under real-life conditions [30] or in experimental settings [57] that dissect the warning process (i.e., perceiving, processing, and complying with app-based warnings). Therefore, the intuitive affective processing of weather warnings poses unanswered questions. To close this gap, the present study introduces the virtual city framework and aims to examine affective processing of a thunderstorm warning message via an experimental, repeated measures design. This virtual city framework is embedded in a browser-based online survey platform [58] that provides easy access and ensures functionality across different browsers and devices. While this increases the reach and applicability of the virtual city framework, it also means lower realism than more complex and immersive virtual environments, such as virtual reality [59] that require additional equiment. In virtual reality, users can create photorealistic representations of real buildings and topographic structures, e.g., [60,61] that can increase immersive effects and perceived realism of the virtual environment. Virtual reality is already widely implemented in (geo-)spatial sciences [60] and construction [62] and it also holds promise for sociobehavioral spatial research, for instance, in evaluating warning compliance under realistic conditions (e.g., [63,64]).

Highly immersive environments can lead to stronger emotional (e.g., negative affective appraisal) and cognitive reactions (e.g., perceived severity, self-efficacy) [65,66] and the evoked reactions might be closer to their real-life equivalent than in low immersive environments. So far, however, findings regarding the ecological validity of these methods (e.g., the congruency between virtual and real behavior) are mixed (e.g., [67,68,69,70]). Hence, more research is needed to examine biopsychosocial processes underlying decision-making in these situations to explore facilitators and barriers to compliance and implementing adaptive behavior when confronted with warnings or hazards.

Therefore, in this experiment, we choose a simpler, less immersive approach where warning message and thunderstorm are presented in two different formats, namely in videos or vignettes. Videos can elicit stronger negative affective appraisals than vignettes, have a higher criterion validity, and they are connected to stronger stimulation of cognitive learning processes [71,72,73,74]. Presumably, videos are able to present a more accurate depiction of contextual cues that increase the perceived realism of a scenario and thus increases external validity [72,74]. Due to their efficiency and high internal validity, however, experimental vignette studies are still popular in applied warning research [29,75,76]. Therefore, both formats will be compared in the current study: Participants will receive a thunderstorm warning message via a mobile app during a virtual city tour, followed by an exposure (vs. non-exposure) to a thunderstorm. Based on previous research, we propose the following hypotheses:

**Hypothesis** **1** **(H1).***The receipt of a warning message regarding a thunderstorm will lead to an increase in momentary anxiety*.

**Hypothesis** **2** **(H2).***The exposure to a thunderstorm will lead to an increase in momentary anxiety*.

**Hypothesis** **3** **(H3).***Presentation format will influence momentary anxiety, in that video footage will lead to greater momentary anxiety than vignettes*.

To enhance the relevance of our research, the present study will additionally focus on the recipient’s search for information in association with momentary anxiety. The search for further information can be part of every stage of the PADM, for example, by inquiring about effective protective measures or suitable locations to seek shelter [20]. Also, individuals who have received a warning message often try to verify and confirm it by seeking out other sources [77]. Such behaviors are connected to negative affect and might be associated with risk and coping appraisals or uncertainty, for instance, regarding message content or behavioral recommendations [78]. Thus, searching for information following a warning message may reduce negative affect by leading to more positive coping appraisals or reducing perceived threat and uncertainty [29,78,79], but it could also be a sign of milling. Milling refers to information seeking behavior following a warning, for example, by searching on other channels and exchanging information with others, which leads to a time delay and further protective measures not being initiated immediately [80]. Hence, information seeking depicts a protective behavior that cannot in itself be labeled as an adaptive or maladaptive response to a warning message, but we can assume that it is predicted by negative affect. Therefore, we will examine the following hypotheses:

**Hypothesis** **4** **(H4).***Higher momentary anxiety will lead to an increased search for further information*.

**Hypothesis** **5** **(H5).***In participants who were confronted with a thunderstorm, the receipt of a warning message, and the presentation format will interact with momentary anxiety, which will lead to an increased search for further information*.

Finally, as already outlined in the PADM, individual characteristics can impact the processing of weather warning messages. Prior experience with severe weather as well as experienced personal harm impact subsequent appraisal of risk and the implementation of preparedness behavior [81,82,83,84,85,86]. Sociodemographic variables, such as age or gender, play a role in risk appraisal in connection with weather warnings and protective measures: female gender and higher age are factors that are associated with higher compliance [12,87,88]. Therefore, age, gender, previous experience with thunderstorms, and storm fear are included as covariates. We will also include current the usage of warning apps as a covariate.

## 2. Materials and Methods

### 2.1. Sample

Data collection took place from February to June 2020. A German adult sample was recruited via various online services (e.g., internet forum posts via Facebook, and platforms for psychological online surveys). Participants were invited to participate in an online study evaluating a novel app as a cover story (see 2.4). Participants were included if they were at least 18 years of age and fluent in German. As an incentive, participants entered a raffle for six gift vouchers (15 Euros).

### 2.2. Materials

The videos used in this study took the viewer on a virtual city tour to Berlin. The virtual scenery was created with the game engine Unity (Unity 5.4.3f1 (64-bit), Unity Technologies, San Francisco, USA, https://unity.com/ (accessed on: 2 August 2021))). For the generation of the videos, Open Broadcaster Software (OBS Studio Version 25.0.8 (64-bit), Hugh Bailey, https://obsproject.com/ (accessed on: 2 August 2021))) and OpenShot Video Editor (OpenShot 2.4.4, Jonathan Thomas, https://www.openshot.org/ (accessed on: 2 August 2021)) were used. Vignettes consisted of screenshots of the videos and written descriptions. Videos and vignettes used in this study can be found in the Appendix A online (File S1 (vignettes) and File S2 (videos)). In both files, videos and vignettes are labelled according to their position in the experiment (1 baseline, 2 warning, 3 no warning, 4 thunderstorm, 5 no thunderstorm).

In this study, a thunderstorm warning message was sent via a simulated mobile phone app. The message was issued by the Deutscher Wetterdienst (DWD). The DWD operates as the German National Meteorological Service and is in charge of the forecast as well as the warning of weather hazards [17]. The warning message contained a description of a pending thunderstorm, possible impact, recommended protective actions, and the affected region. The message was adapted from previously issued severe weather warnings, optimized for mobile warning systems based on current recommendations [20,54,89]. The thresholds for severe weather were based on DWD criteria for warnings of severe weather (so-called Level 3 events) [90]. The original message (in German) and its English translation can be found in Figure 1. The full text is also available upon request from the first author.

### 2.3. Measures

Momentary anxiety was assessed with one item (“How anxious do you feel at the moment?”) at three time points. A 5-point Likert scale was used, ranging from 1 (not at all) to 5 (extremely). The search for further information was assessed using one item (“How likely is it that you would search for more information about this situation?”). Again, a 5-point Likert scale was used, ranging from 1 (not at all) to 5 (very likely).

Seven items were used to measure previous thunderstorm experience with 7-point Likert scales (1 (never) to 7 (very often)). For example, participants were asked whether they or their friends and family had already experienced a thunderstorm, or whether they were in danger during a thunderstorm. To measure storm fear, the Storm Fear Questionnaire (SFQ) [47] was used. The SFQ consists of 15 items that address weather- and storm-related phobia in adults. Current warning app use was assessed via one item (“Do you currently use one or more apps to be warned of weather events?”); a ‘yes’ indicated current app use. Gender (1 = male, 2 = female) and age (in years) were assessed using single items.

### 2.4. Study Design and Study Procedure

Participants were invited via Facebook posts, and online survey platforms (e.g., Psychology Today https://www.psychologie-heute.de/aktuelles/studienteilnahme.html (accessed on: 2 August 2021)) to test and rate a touristic app that aims to provide multifaceted information about location, sights, traffic, and weather. A hyperlink directed participants to the experiment, which was implemented via SoSci Survey, Version 3.2.33 [58] and took about 30 minutes to complete. Participants were welcomed to the study and received information about the study background and how their data would be collected and used. After they gave their informed consent, participants first reported sociodemographic data (e.g., age and gender). Then, they received a cover story asking them to take part in the evaluation of a new touristic city app called “Local App” during a virtual city tour.

Data collection took place using a quasi-experimental 2 × 2 × 2 factorial design. The study procedure can be found in Figure 2. Due to randomization, a subset of participants received all further footage of the virtual city either via videos or vignettes (video/vignette). Following randomization by random draw, momentary anxiety was assessed for the first time (t1). Then, one group received a thunderstorm warning message while the other group received neutral information about the city (warning message/no warning message), followed by the second assessment of momentary anxiety (t2). Finally, either a thunderstorm or neutral information was presented (thunderstorm/no thunderstorm), and momentary anxiety was assessed for the last time point (t3).

Lastly, prior thunderstorm experience, storm fear, and current use of warning apps were assessed in a questionnaire. Upon completion, all participants were given information for psychosocial support services if they needed assistance due to the exposure to a thunderstorm. The required sample size for this study was estimated a priori using G × Power [91], based on an anticipated power of β = 0.80, an alpha error value of α = 0.05, a medium effect size of f = 0.25, eight groups, and three repeated measurements with an estimated non-correlation between repeated measures (due to experimental manipulation). An ANOVA with repeated measures, and within-between interactions was chosen as the statistical analysis, resulting in a required sample size of *n* = 104.

### 2.5. Statistical Analysis

All analyses were performed using IBM SPSS Statistics for Windows, Version 27.0 (IBM Corp., Armonk, NY, USA) and PROCESS macro, Version 3.0 (https://onlinelibrary.wiley.com/doi/abs/10.1111/jedm.12050 (accessed on: 2 August 2021)). Means and standard deviations were calculated for the SFQ and prior thunderstorm experience. Chi-square tests and t tests were performed to analyze differences between experimental conditions. Pearson correlations were performed to investigate links between the examined constructs.

To answer Hypotheses 1 to 3, a repeated measures ANOVA as well as ANCOVA were conducted with time (i.e., momentary anxiety at three time points) as a within-person factor and warning, thunderstorm, and format (video/vignette) as a between-person factors. As a measure of effect size, partial eta squared (*η_p_*²) was reported. To correct violations of sphericity, Greenhouse-Geisser adjustment was applied. To answer Hypotheses 4 and 5, a moderation analysis was conducted via multiple linear regression. For the repeated measure ANCOVA and moderation analysis, age, gender, prior thunderstorm experience, storm fear, and app use were included as covariates. The moderation analysis was performed in the subsample of participants who were confronted with a thunderstorm during the experiment (*n* = 209).

## 3. Results

### 3.1. Descriptive Statistics

A total of 276 participants completed the survey. Missing values on item level ranged from 0.0% (e.g., momentary anxiety at t1) to 5.4% (app use). Most participants had no or one missing value (89.3%), followed by multiple missing values in previous experience of thunderstorms (3.7%), and storm fear (3.7%). Complete cases analysis was performed.

The sample included 62.0% females (*n* = 171) and 34.1% males (*n* = 94), with no information received from 4.0% (*n* = 11). Participants’ age ranged from 17 to 83 years (*M* = 41.07, *SD* = 16.44). Mean score of previous thunderstorm experience was 2.90 (*SD* = 0.71, Range = 1.00–6.43) and mean score of storm fear was 1.50 (*SD* = 0.54, Range = 1–5). About 31.2% (*n* = 86) of participants reported current use of a mobile warning app for weather events.

According to the randomization, 141 (48.9%) participants received a warning message, 209 (75.7%) participants were confronted with a thunderstorm, and 146 (52.9%) received their information as videos. The thunderstorm condition was oversampled to account for potential dropout due to technical difficulties in playing the videos that were reported in a pilot test of the experiment. The sample size for each group (*n* = 12–56) as well as descriptive statistics for each group can be found in Appendix B (Table A1). The eight groups did not differ by age, gender, previous thunderstorm experience, storm fear, and app use, pointing to a successful randomization procedure. The only difference was observed for format and gender (*χ*²(1) = 8.15, *p* < 0.01), in that more female participants received information via vignettes.

Bivariate correlations of study variables can be found in Table 1. Significant positive correlations were found for momentary anxiety (t1–t3) and the search for further information (t3) (*r* = 0.13–0.31, *p* < 0.05–0.001). In the warning message group, significant positive associations were found for momentary anxiety at t2 (*r* = 0.34, *p* < 0.001) and t3 (*r* = 0.14, *p* < 0.05), while exposure to a virtual thunderstormwas positively correlated with momentary anxiety at t3 (*r* = 0.34, *p* < 0.001). Thus, receiving a warning message and being exposed to a thunderstorm were associated with higher anxiety ratings. Negative correlations were observed for presentation format and momentary anxiety at t2 (*r* = −0.12, *p* < 0.05) and t3 (*r* = −0.14, *p* < 0.05), meaning that vignettes caused less momentary anxiety than videos.

### 3.2. Trajectory of Momentary Anxiety

The following section presents the results of the ANCOVA regarding momentary anxiety (H1 to H3), and the regression of information seeking intentions on momentary anxiety (H4 and H5). A repeated measures ANOVA without covariates showed three significant effects: a main effect of time, indicating an increase in momentary anxiety over time (*F*(1.81, 482.64) = 34.148, *p* < 0.001, *η_p_*² = 0.113), an interaction of time x warning (*F*(1.81, 482.64) = 8.19, *p* < 0.05, *η_p_*² = 0.03), and an interaction of time x thunderstorm (*F*(1.82, 482.64) = 27.64, *p* < 0.001, *η_p_*² = 0.09), with participants receiving a warning or being exposed to a thunderstorm reporting higher anxiety scores. The remaining effects did not reach statistical significance.

**H1.** *The receipt of a warning message regarding a thunderstorm will lead to an increase in momentary anxiety*.

Repeated measure ANCOVA revealed a significant interaction of time and receiving a warning message (*F*(1.82, 412.01) = 7.84, *p* < 0.01, *η_p_*² = 0.03). Participants who received a warning message regarding a thunderstorm reported higher momentary anxiety compared to participants who received no warning.

**H2.** *The exposure to a thunderstorm will lead to an increase in momentary anxiety*.

A significant interaction was found for time and exposure to a thunderstorm (*F*(1.82, 412.01) = 21.70, *p* < 0.001, *η_p_*² = 0.09), in that participants who were confronted reported higher momentary anxiety. Trajectories of momentary anxiety divided for warning receipt and thunderstorm exposure can be found in Figure 3.

**H3.** *Presentation format will influence momentary anxiety, in that video footage will cause more momentary anxiety than vignettes*.

No significant interaction was found for time and presentation format (*F*(1.82, 412.01) = 0.59, *p* = 0.54, *η_p_*² = 0.00).

### 3.3. Association between Momentary Anxiety and Information Seeking

Hypotheses 4 and 5 examined the association between momentary anxiety (t3) and information seeking intentions via hierarchical regression models.

**H4.** *Higher momentary anxiety will lead to an increased search for further information*.

Multiple regression analysis for participants who were confronted with a thunderstorm revealed a significant overall model (*F*(10.00, 171.00) = 2.83, *p* < 0.01, R² = 0.12). Momentary anxiety (t3) predicted the search for further information (b = 0.30, *p* < 0.001, 95% CI = 0.14–0.47), while the other effects were non-significant.

**H5.** *In participants who were exposed to a thunderstorm, the receipt of a warning message, and the presentation format will interact with momentary anxiety, which will lead to an increased search for further information*.

The receipt of a warning message (b = 0.25, *p* = 0.10, 95% CI = −0.05–0.56, Δ*R*² = 0.01, *F*(1.00, 171.00) = 2.66) and presentation format (b = −0.07, *p* = 0.65, 95% CI = −0.38–0.24, Δ*R*² = 0.00, *F*(1.00, 171.00) = 0.20) did not significantly moderate the effect between momentary anxiety (t3) and search for further information.

Regarding the examined covariates, a significant interaction was found for momentary anxiety and age in the ANCOVA (*F*(1.82, 412.01) = 3.20, *p* < 0.05, *η_p_*² = 0.01). Higher age was associated with less momentary anxiety at the first point of measurement (t1). For the remaining covariates, no significant interactions were found.

## 4. Discussion

The present study investigated changes of momentary anxiety and intention to seek further information following the receipt of an app-based thunderstorm warning message, and the exposure to a thunderstorm in a virtual city. Results showed that receiving a warning message and being confronted with a virtual thunderstorm both significantly increased momentary anxiety compared to control conditions, which in turn predicted intentions to seek further information. However, neither format (video or vignette) nor the interaction between warning and thunderstorm had a significant impact on momentary anxiety.

### 4.1. The Role of Affective Appraisal in App-Based Warning Processes

Statistical analyses confirmed Hypotheses 1 and 2, in that momentary anxiety increased after receiving a warning message, as well as after being confronted with a thunderstorm. These trajectories point to the activation of the experiential system as proposed by the affect heuristic [36,37,38], and the behavioral inhibition system as defined by the BIS/BAS model [39,40,41,42]. Facing an unexpected or potentially threatening stimulus (in the present case, a warning of an impending thunderstorm) appeared to activate appropriate systems and induce affective reactions. This observation corroborates previous research on the processing of severe weather hazards [47,48,49]. In line with fear appeal theory [33,34], and the PADM [21], this affective appraisal could indicate enhanced threat appraisal.

In contrast to the first two hypotheses, Hypothesis 3 was not confirmed. This means that there was no difference in the reported level of momentary anxiety regarding the presentation format. Videos, as well as vignettes, seem to be equally suitable for the experimental investigation of thunderstorm warning messages. This contrasts previous studies [71,72,73,74] that pointed to stronger affective reactions to videos. However, in said studies, affect was mediated by perceived realism of the scenario, which we did not assess in this study, therefore we could not compare the perceived realism of both formats. Since higher perceived realism is associated with stronger affective reactions [65,66], this could be a limiting factor of our study, because neither videos nor vignettes are highly immersive, unlike modern virtual reality applications [92,93].

However, the association between perceived realism and anxiety is more complex: research on the connection between perceived realism (as a part of the overall sense of presence in a virtual environment) and emotional arousal (e.g., anxiety) suggests that exteroception (i.e., immersive qualities of the environments) as well as interoception (i.e., physiological arousal) influence assessments of realism and anxiety [94]. Since we observed a significant increase in anxiety ratings, we assume that it was not strongly affected by the immersiveness of the technology, and thus exteroceptive cues, but rather the intero-ceptive appraisal of the threat.

In future research, a direct comparison of high immersive technology (e.g., virtual reality) and low immersive technology (e.g., vignettes) implementing the virtual city framework can provide more information on the interplay of immersion, and perceived realism on affective and cognitive processing of warning messages throughout the warning process.

Furthermore, qualitative research methods can be integrated to identify relevant dimensions of perceived realism to enhance immersiveness and arousal. A qualitative analysis of a virtual reality fire extinguisher training, for example [95], revealed specific aspects of the virtual environment that were connected to perceived realism and could be addressed in future iterations of the training to improve training effects.

Finally, field studies promise higher validity and allow for in-situ assessments of physiological reactions in addition to cognitive and affective appraisals of warning messages (e.g., via ambulatory assessments) [11]. A recent study on app-based weather warnings [96] found that affective and cognitive but not behavioral responses to warning messages differed between experimental and field approaches. In the field experiment, participants reported better understanding of warning message, threat and how to respond, but also less concern regarding one’s safety. But since the research was cross-sectional and did not examine cognitive and affective appraisal of warning messages prospectively, further research is necessary to examine and compare trajectories of appraisal and subsequent behaviors in field studies as well as experiments. Combining field studies and the experimental approach of the virtual city framework could be a promising area for future warning research.

### 4.2. Information Seeking versus Milling

By examining the search for further information, the present study mapped another step of processing warning messages as described in the PADM, namely behavioral responses [20,21,22]. Hypothesis 4 was confirmed as higher momentary anxiety at the third point of measurement was associated with increased intention to search for more information. The results are in line with prior research, which found that emotional reactions, such as fear, towards real-time weather events could have a main impact on the implementation of behavioral responses [30]. Activated by the BIS [39,40,41,42], momentary anxiety could interrupt the current activity and promote seeking information for a better orientation. This interpretation was further supported by testing Hypothesis 5, which revealed that warning message receipt did not affect the intention to seek further information, thus Hypothesis 5 had to be rejected. This indicates that momentary affect is particularly important in the decision-making process following warning messages or weather events, which corroborates the affect heuristic [38]. Moreover, our findings contradict the phenomenon of milling. According to milling, the receipt of a warning message leads to an increased search for information as the individuals at risk do not feel sufficiently informed and want to exchange information with each other before taking protective measures [80]. In this study, although momentary anxiety increased with warning message receipt, it also decreased when no threat occurred. Thus, the search for further information seems to be affected by the thunderstorm itself rather than by the warning message. This, is evident in the non-significant correlative relationship between message receipt and information search.

Nevertheless, the search for further information embodies only one (protective) behavior, which can be both beneficial and detrimental [80]. Also, for most hazards as well as for thunderstorms, a combination of several measures needs to be implemented to protect oneself (e.g., closing doors and windows, search for shelter). Future studies should examine all protective measures conveyed in a warning message. In this context, the intention to act is suitable, as is the actual implementation of protective measures. Field studies could allow situational analyses by using ambulatory assessments [96].

In an effort to include characteristics of the warning message recipients, statistical analyses were controlled for age, gender, previous thunderstorm experience, app use, and storm fear. With one exception (i.e., a negative association between age and anxiety at baseline), none of these variables affected momentary anxiety. However, the bivariate correlations (Table 1) showed that the investigated covariates were associated with anxiety, and they may still become relevant in other stages of the warning process. For example, participants who were currently using a weather warning app showed less momentary anxiety at the second point of measurement (i.e., when they received a warning). This may indicate that the affective reaction was reduced due to existing experience with such warning messages, which may have implications for threat and efficacy appraisals within the compliance process [34], as previous research pointed to a connection between positive affect (i.e., less negative affect), and lower intentions regarding behavior change, for instance, towards pro-environmental behavior [35]. This interplay of warning app use, affective and cognitive risk appraisal, and compliance warrants further research.

Positive correlations also emerged between app use and prior thunderstorm experience, as well as app use and storm fear. One can assume that individuals who had already experienced severe weather and were perhaps harmed were more likely to develop a fear regarding storms and, therefore, wanted to protect themselves through the use of weather warning apps [81,83,84,85]. Although these correlations do not imply causality and require further investigation, it becomes apparent that the characteristics of the message receivers should not be neglected.

### 4.3. Limitations

The present study clearly has limitations that should be considered when interpreting the results. The results are based on a German convenience sample. Therefore, no representative statements can be made. Since weather-related fears are relatively low in Germany and European countries vary in their risk communication practice and policies [9,24], an international comparison would be of interest.

Although this study was based on established and well-known definitions of severe weather (according to the DWD), and recommendations for designing mobile public warning messages [11,20,89], we did not examine the realism and perceived of each vignette or video. Prior research showed that the more severe or threatening a weather event is perceived, the more likely it is for those at risk to take protective actions and start to implement them at an earlier stage [83]. Because of this, future research should consider various levels of severity, and assess cognitive and affective appraisals of the hazard simultaneously. Similarly, effects of differing warning message content and style could also be tested, as we were not able to vary aspects, such as length and design, that may have an impact on affective reactions [78].

Although the participants in the present study were presented with realistic footage of warning messages sent by an app and thunderstorms (e.g., video material), we were not able to collect data under real-life conditions (e.g., field studies). Recent research shows, however, that scenario-based surveys do not seem to differ from field assessments in their results [30]. Nevertheless, the virtual city framework could also be tested in more realistic conditions, for example, by using virtual reality [64], and be extended by more fine-grained assessments, such as ambulatory assessments to further explore psychosocial mechanisms of warning compliance and observe behavioral reactions.

## 5. Conclusions

Overall, the results of this study indicate that app-based warning messages regarding a severe weather event as well as the event itself differentially affect momentary anxiety (i.e., negative affect) throughout the warning process, when controlling for previous warning app use as well as experience with thunderstorms. While the study has methodological limitations, it provides support for a virtual city framework to test warning messages and subsequent appraisals, and potentially behaviors. The multi-step structure allows for a segmentation of the warning process that can be used to study psychophysiological, cognitive, and affective correlates of each step of the warning process more closely. With mobile warning apps increasing in popularity and reach, the digital setting presented in this study can be useful to develop and test different types of warnings and conditions with presumably high internal validity, before implementing them in field studies, and testing their ecological validity. The set-up could also be used to implement warning-based trainings, for senders and receivers of warning messages. Therefore, we welcome future research to replicate and refine the virtual city framework in more diverse samples, using different measures and investigating different hazards, to improve our understanding of app-based warning processing, and strengthen preventive efforts and resiliency.

## Figures and Tables

**Figure 1 ijerph-18-08353-f001:**
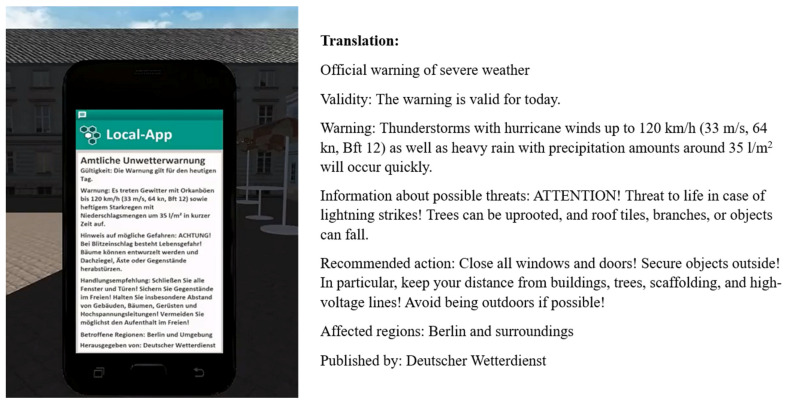
Warning message used at second point of measurement (t2; in German), and English translation of the warning message.

**Figure 2 ijerph-18-08353-f002:**
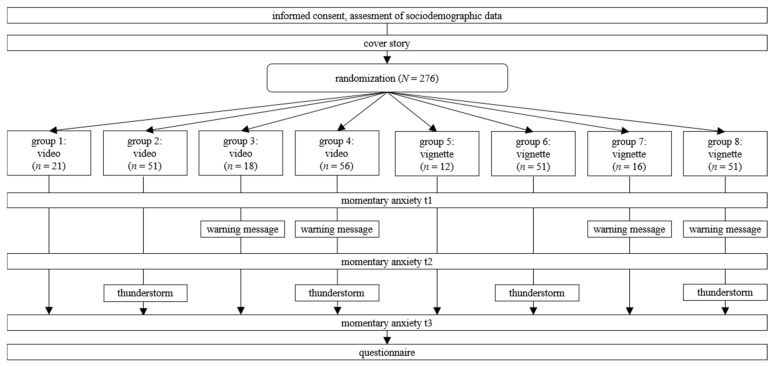
Study procedure: Participants were randomized into eight groups according to presentation format (video vs. vignette), warning receipt (no warning vs. warning), and thunderstorm exposure (no thunderstorm vs. thunderstorm). Momentary anxiety was assessed at three time points: t1 (beginning of the cover story), t2 (warning message receipt vs. no receipt), t3 (thunderstorm exposure vs. no exposure).

**Figure 3 ijerph-18-08353-f003:**
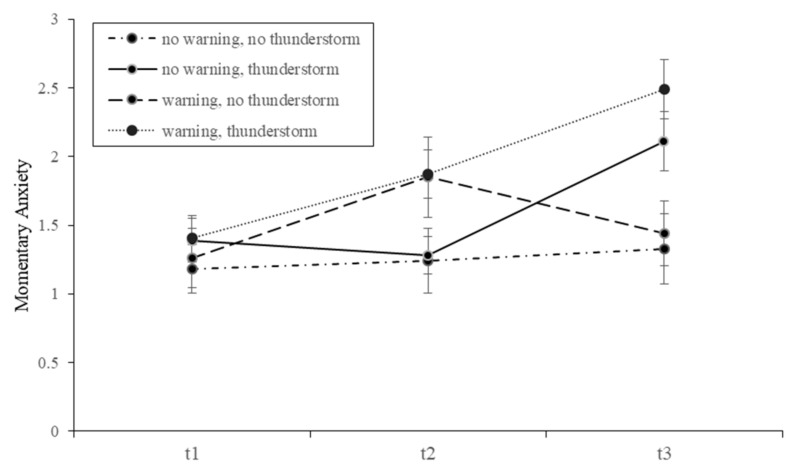
Mean values of momentary anxiety over the course of the study. Note. The lines represent four groups (warning/no warning x thunderstorm/no thunderstorm); presentation format was not included as an additional factor due to a lack of statistical significance in the ANOVA and ANCOVA. Points of measurement: t1 (beginning of the cover story), t2 (warning message receipt vs. no receipt), t3 (thunderstorm exposure vs. no exposure). Momentary anxiety was assessed using a 5-point Likert scale, ranging from 1 (not at all) to 5 (extremely). Error bars show the 95% confidence interval.

**Table 1 ijerph-18-08353-t001:** Pairwise (Pearson) correlations of momentary anxiety (t1–t3), search for further information, warning message receipt, thunderstorm confrontation, and presentation fomat, including covariates (age, gender, thunderstorm experience, storm fear, app use), *n* = 250–276.

		1	2	3	4	5	6	7	8	9	10	11	12
**1**	momentary anxiety t1	1											
**2**	momentary anxiety t2	0.49 ***	1										
**3**	momentary anxiety t3	0.39 ***	0.53 ***	1									
**4**	search for further information	0.13 *	0.25 ***	0.31 ***	1								
**5**	warning message receipt	0.02	0.34 ***	0.14 *	0.10	1							
**6**	thunderstorm exposure	0.10	0.02	0.34 ***	0.18 **	0.00	1						
**7**	format	−0.01	−0.12 *	−0.14 *	0.03	−0.01	−0.06	1					
**8**	age	−0.12 *	−0.09	0.02	0.08	−0.05	0.13 *	0.02	1				
**9**	gender	0.07	0.11	0.14 *	0.12 *	0.02	0.05	−0.18 **	−0.18 **	1			
**10**	thunderstorm experience	0.11	0.01	−0.02	0.08	0.02	−0.03	−0.07	0.08	−0.04	1		
**11**	storm fear	0.20 **	0.23 ***	0.19 **	0.12 *	0.01	−0.02	−0.01	−0.02	0.01	0.23 ***	1	
**12**	app use	−0.10	−0.12 *	−0.05	0.02	0.11	0.09	0.01	0.21 **	−0.09	0.15 *	0.18 **	1

Note. t1–t3: Points of measurement: t1 (beginning of the cover story), t2 (warning message receipt vs. no receipt), t3 (thunderstorm exposure vs. no exposure). Momentary anxiety was assessed using a 5-point Likert scale, ranging from 1 (not at all) to 5 (extremely). Warning message receipt (0 = no warning. 1 = warning); thunderstorm exposure (0 = no exposure. 1 = exposure); presentation format (1 = vignette. 2 = film); gender (0 = male, 1 = female); app use (0 = no current use. 1 = current use). * *p* < 0.05. ** *p* < 0.01. *** *p* < 0.001.

## Data Availability

The data presented in this study are available on request from the corresponding author. The data are not publicly available due to ethical concerns.

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
