# Peer review of "A Walk in the Park? Examining the Impact of App-Based Weather Warnings on Affective Reactions and the Search for Information in a Virtual City"

_ijerph, 2021, doi:10.3390/ijerph18168353_

Round 1

Reviewer 1 Report

It seems to be an interesting study, introducing a virtual city framework to examine reactions when receiving an app-based warning, and subsequent behavioral intentions.

  1. For the participants data, you can add a description to explain its nature; or increase the number of samples;
  2. Add some psychological or emotional opinions and evidence in the experiment part or discussion part?
  3. Can it be combined with some actual investigations?
  4. Add some definitions of severe weather.
  5. Some paragraphs can be combined;
  6. Some spelling or formatting errors, please check.

Author Response

Dear Reviewer,

We thank you for the opportunity to revise and resubmit our manuscript. The revisions and have improved our manuscript and connected it more strongly to previous research. We have checked the manuscript for spelling and formatting errors; our changes are tracked in the text. We hope that you are satisfied with the changes we made. Please see the attachment for our point-by-point responses to your comments.

Sincerely,
Samuel Tomczyk, on behalf of all authors

Reviewer 2 Report

The manuscript is based on a timely research question and a solid literature body. I would like to mention these two points which the author team should consider for a revised version of the manuscript:

  • The beginning of the manuscript should be extended, as the possibilities of modern virtual reality applications addressing (geo-)spatial phenomena go far beyond meteorological impacts. In your study, you use a “virtual city”, but not an immersive application. Nevertheless, it would be worth pointing to the possibilities of the modern creation of virtual environments. The creation of immersive virtual environments is under study, and in the last three years, many examples were published where open (public) data initiatives were used to create realistic 3D representations (incl. photorealistic buildings and representations of topographic / relief structures). [For e.g. https://doi.org/10.1007/s42489-020-00069-6 and https://doi.org/10.1007/s41064-020-00106-z and https://doi.org/10.1080/10095020.2019.1621544 ]. Those developments could help to even increase the levels of realism and immersion. Spatially oriented problems could benefit from these solutions. Moreover, there are numerous studies available which discuss possible applications of immersive virtual environments, such as urban planning (https://doi.org/10.1016/j.cities.2019.102559), cultural heritage (https://doi.org/10.1007/s41064-020-00091-3) and noise pollution (https://doi.org/10.3390/mti3020034 ). Indicating a broader field of spatial application and construction opportunities of realistic immersive virtual environments could help to improve the introduction.

  • The aspect of anxiety is a key part of the study. Anxiety has individual strength and is likely dependent on the degree of immersion. In other words, if participants are not fully immersed into a virtual environment (and you do not operate with an ‘immersive virtual environment’), anxiety might well differ and have a strong individual impact on your study result. How did you face this problem?

Author Response

(The authors gave the same response as above.)

Round 2

Reviewer 1 Report

In this version of the paper, most of  my comments, concerns and suggestions from the review of the previous version were fully answered.  However, I can't see the response to Comment 2.2 by Revewer #2 for responsing my point 2? And, please explain again that which small paragraphs you have combined?
